# A Semi-Supervised Approach to Sentiment Analysis of Tweets during the 2022 Philippine Presidential Election

Julio Jerison E. Macrohon [1,*], Charlyn Nayve Villavicencio [1,2], X. Alphonse Inbaraj [1] and Jyh-Horng Jeng [1]

1. Department of Information Engineering, I-Shou University, Kaohsiung City 84001, Taiwan
2. College of Information and Communications Technology, Bulacan State University, Bulacan 3000, Philippines
* Correspondence: isu10903050d@cloud.isu.edu.tw

**Abstract:** With the increasing popularity of Twitter as both a social media platform and a data source for companies, decision makers, advertisers, and even researchers alike, data have been so massive that manual labeling is no longer feasible. This research uses a semi-supervised approach to sentiment analysis of both English and Tagalog tweets using a base classifier. In this study involving the Philippines, where social media played a central role in the campaign of both candidates, the tweets during the widely contested race between the son of the Philippines' former President and Dictator, and the outgoing Vice President of the Philippines were used. Using Natural Language Processing techniques, these tweets were annotated, processed, and trained to classify both English and Tagalog tweets into three polarities: positive, neutral, and negative. Through the Self-Training with Multinomial Naïve Bayes as base classifier with 30% unlabeled data, the results yielded an accuracy of 84.83%, which outweighs other studies using Twitter data from the Philippines.

**Keywords:** 2022 Philippine Presidential Election; semi-supervised learning; Natural Language Processing; sentiment analysis; Python; social media; Twitter; tweets





## 1. Introduction

The 2022 Presidential Election in the Philippines has been a hotly contested race between the son of the Philippines' former President, Dictator, and namesake, Ferdinand "Bong Bong" R. Marcos, Junior, and the outgoing Vice President of the Philippines, Ma. Leonor "Leni" G. Robredo.

The Philippines holds its Presidential and Vice-Presidential Elections every six (6) years where both the President and Vice President are elected separately [1]. Over the past few years, both the elected President and Vice President came from two opposing parties [2]. During the last Presidential and Vice-Presidential Elections in 2016, both Marcos, Jr. and Robredo were vying for the position of Vice President, of which the winner was the latter, despite electoral protests from the former [3].

Despite losing the Vice Presidency, the 2022 Presidential election saw Marcos, Jr. win in a landslide victory taking more than 30 million (about 58%) or majority of the votes. His running mate, outgoing President Rodrigo R. Duterte's daughter, former Davao City Mayor Sara Z. Duterte-Carpio also won an overwhelming majority or 32 million votes (roughly 61%) [4].

As with most elections, the 2022 Presidential election was regarded as one of the most divisive elections in the Philippines [5]. The purpose of this study is to analyze the sentiment of Filipinos regardless of whom they support. According to StatCounter, Twitter is the third most used social media Site in the Philippines [6], which was hailed as the "World's Social Media Capital" [7].

Fortunately, most of the people who post on Twitter are supporters of the top two (2) candidates for President and also Vice President, the tandem of Marcos, Jr. and Duterte-Carpio, and the Robredo-Pangilinan tandem. Thus, the researchers used keywords to

collect the tweets only pertaining to the two tandems. Other candidates for this years' elections including Manila Mayor Francisco "Isko Moreno" Domagoso, Senator Panfilo "Ping" Lacson, famed boxer Senator Emmanuel "Manny" Pacquiao, and other candidates and their respective running mates were included in the gathering, but due to a limited number of relevant tweets, the researchers decided to focus on the top two candidates instead. In this regard, the researchers collected tweets from 8 February to 8 May 2022, which the Philippine Commission on Elections (COMELEC) promulgated as the campaign period for the national candidates.

### 1.1. Contributions of this Paper

The main contributions of this study can be summarized as follows:

- Automatic labeling of polarity of English and Filipino/Tagalog language tweets.
- Reporting of the sentiment of the public toward either or both of the candidates.
- Using the study as a topic in Philippine sociology or political science.
- Using the proposed model to further analyze tweets for the next general elections.
- Help advance the NLP research and practice in Filipino/Tagalog language.

### 1.2. Organization of This Paper

In this paper, the researchers compared the methodology to that of other papers as seen in Section 2. Then, explained the step-by-step approach in pre-processing and processing of the data gathered in Section 3, of which the results were then displayed and discussed in Section 4 together with the researchers' conclusions in Section 5.

## 2. The Related Literature

In the Web 2.0 age, social media has been a widely used tool to air sentiments, views, and opinions about each and every topic in the world today. It is fueled by the ease of use and accessibility of the internet and the media such as mobile phones, tablets, and laptops [8]. Use of these social media sites has been a go-to for users who need some kind of outlet to air their sentiments. Of all the social media sites, no one compares to that of Twitter. Though Facebook has a bigger user base, Twitter users can simply post or "tweet" up to 280 characters (previously 140 characters) of whatever they want. Boasting a user base of 100 million daily active users and 500 million tweets sent daily, Twitter has grown exponentially from just an SMS-based platform to spreading information fast [9].

### 2.1. Sentiment Analysis in the Context of Elections

Election seasons have been widely researched in the academic community. Specifically pertaining to the context of its effects on society. In the study of Bansal et al., they introduced a novel method called Hybrid Topic Based Sentiment Analysis (HTBSA) that aimed at capturing word relations and co-occurrences in the Uttar Pradesh (India) election-related tweets [10]. For the Indian General Elections in 2019, Sharma et al. used Twitter to gather data and used R for the pre-processing and analysis, which was confirmed by the actual election results with the candidate winning said election [11]. In the 2019 Spanish elections, Rodríguez-Ibáñez et al. applied a two-fold analysis strategy using temporal analysis and intrinsic embeddings [12]. During the 2016 US Presidential Elections, sentiment analysis was performed for Russian IRA troll messages which concluded positive sentiments toward Donald Trump and negative sentiments for Hilary Clinton which benefited the former [13]. A study by Bansal et al. added Emojis and n-Gram features for vote share prediction in the 2017 Uttar Pradesh (India) legislative elections which significantly decreased prediction error [14]. Spatio-temporal trend analysis was also performed in the 2014 Brazilian Presidential Election [15], having close to 90% accuracy using SVM algorithm for classification. Spatio-temporal trend analysis was also conducted in the study of Kovács et al. on how European Twitter users reacted to the murder of Slovakian journalist, Ján Kuciak, which affected the course of discussions in Europe [16].

Lastly, in the study of Kramer et al., an accuracy of 89.09% was achieved using Naïve Bayes with Minimal Recursion Semantics (MRS)-based features [17].

### 2.2. Sentiment Analysis Using Mixed English and Filipino Language

With several published research papers using Twitter to analyze sentiment of users, unfortunately, they are mostly for the English language. While Filipinos generally tweet in English, some do actually mix in Filipino/Tagalog or even other languages or vernaculars such as Visayan/Bisaya, Ilocano, Ilonggo/Hiligaynon, Zamboangueño/Chavacano, and others. Thereby, researchers in the Philippines have tried to add or mix said languages.

Research from Parde et al. introduced a large dataset of about 20,000 instances for code-switching of Filipino/Tagalog tweets [18]. This will help facilitate future research to better understand and annotate these tweets.

It is evident that this research was conducted to extend that our previous article on COVID-19 vaccine sentiment analysis [19]. Though with different corresponding authors, both first and second authors worked hand in hand in the creation of both articles. A comparison of other relevant research based on the Philippine setting or tweets can be seen in Table 1 below.

**Table 1.** Comparison of Philippine Tweet classification results.

| Authors | Language | Classifier | Accuracy | References |
|---|---|---|---|---|
| Samonte et al. | English | Naïve Bayes | 66.67% | [20] |
| Abisado et al. | English, Tagalog | Multinomial Naïve Bayes | 72.17% | [21] |
| Villavicencio et al. | English, Tagalog | Naïve Bayes | 81.77% | [19] |
| **Proposed Method** | **English, Tagalog** | **Multinomial Naïve Bayes with Self-Training** | **84.83%** | |

The studies of Samonte et al. used RapidMiner to compare three models: Naïve Bayes, support vector machine, and random forest to analyze English tweets about local airlines. After three attempts, Naïve Bayes yielded a result of 66.67% accuracy [20].

Abisado et al. used the same approach with the previous research, but did not use RapidMiner and instead used Python, similar to that of this research. Using Multinomial Naïve Bayes to annotate English and Tagalog sentiments during the COVID-19 pandemic, the study obtained an accuracy of 72.17% [21].

The third paper, which has the same authors as this paper, uses the approach from Samonte et al., this time using the data about COVID-19 vaccines from RapidMiner during the first month when the vaccines became available to the world. Using the Naïve Bayes model, our research yielded an accuracy of 81.77% for English and Tagalog tweets. This paper obtained an 84.83% accuracy using Multinomial Naïve Bayes for both English and Tagalog tweets, which is the highest in terms of accuracy.

## 3. Methodology

An eight-phase approach was used in order for this study to be conducted. The first being tweet collection, then annotation, pre-processing, word embedding, hyperparameter tuning, comparative analysis, semi-supervised learning, and lastly, performance evaluation. This is illustrated in Figure 1.

### 3.1. Data Collection

Through the Python library, *tweepy*, the researchers accessed the Twitter API to download all the tweets by specifying the needed filters. By using PyCharm, tweets from 28 February up to 8 May 2022, were collected due to the fact that these dates were the campaign period, as specified by the Philippine Commission on Election (COMELEC). Queries such as "Bong Bong Marcos", "Leni Robredo", "Sara Duterte", "Kiko Pangilinan", and others were used to make the data relevant. The researchers also included the names of the Vice-Presidential candidates since most tweets involved both the Presidential and Vice-Presidential candidates. Table 2 reflects the raw tweets extracted.

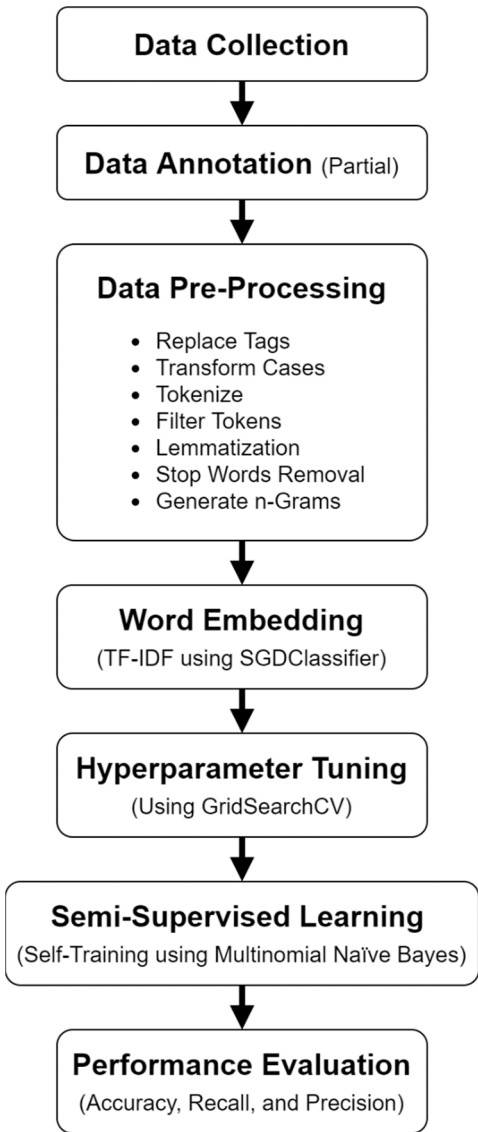

**Figure 1.** Eight Phases of the Study: Data Collection, Data Annotation, Data Pre-Processing, Word Embedding, Hyperparameter Tuning, Comparative Analysis, Semi-Supervised Learning, and Performance Evaluation.

A total of 150,729 raw tweets were collected. Some of which were duplicates and retweets (RTs); thus, the researchers used RapidMiner to remove duplicates, the process of which can be seen in Figure 2. The process further reduced the tweets to be annotated, now totaling 114,851 tweets.

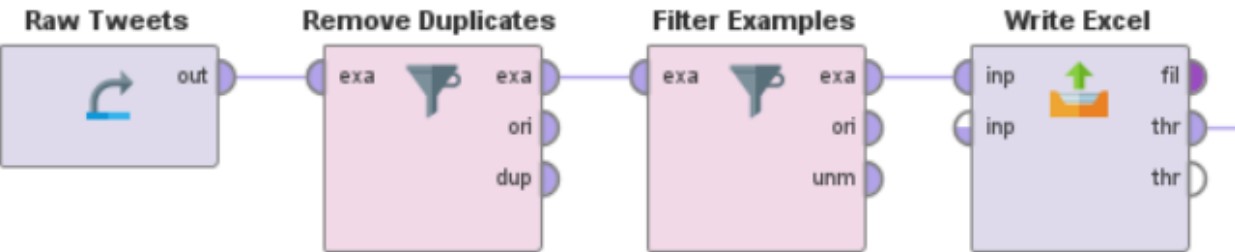

**Figure 2.** Process Flow in RapidMiner from raw tweets of which duplicates were removed and filtered before writing an Excel File.

**Table 2.** Some of the raw tweets extracted from *tweepy*.

| Date of Creation | Tweet |
|---|---|
| 27 February 2022 | #IpanaloNa10To<br>9 out of 10 Presidential candidates agreed to show their SALN.<br>BONG-BONG MARCOS, KAYA MO RIN BA??? |
| 02 March 2022 | OCTA's recent survey showed the Bongbong Marcos-Sara Duterte tandem topping the list of preferred presidential and VP bets |
| 03 March 2022 | Vice President Leni Robredo and her running mate Senator Kiko Pangilinan held a dialogue with farmers and fishermen |
| 18 March 2022 | Celebs endorse bets: Willie Revillame for Sara Duterte, Mocha Uson for Isko Moreno #Halalan2022 |
| 3 April 2022 | "Ang tatanglaw sa buong bayan, ilaw ng tahanan."—VP @lenirobredo<br>#LetLeniLead<br>#10RobredoPresident |
| 18 April 2022 | #BataanIsPink<br>#CatrionaIsPink<br>#HindiIsusukoAngBataan<br>#LetLeniLead |
| 24 April 2022 | @BBMSARAH22 I BELIEVE NA MAS MALAKAS SI SIR BONG BONG MARCOS KESA KAY HESU KRISTO!! LIKE IF YOU AGREE<br>#LeniWithdraw<br>#taguigispulaasf*ck |
| 24 April 2022 | "It's okay to disagree, but don't let your hate overcome your humanity."-BBM<br>#BBMIsMyPresident2022 |
| 28 April 2022 | #1 Trending parin. BBM FOR PRESIDENT<br>PULA ANG PAMPANGGA<br>#PampamBNK48<br>#BBMIsMyPresident2022<br>#BBMSaraUNITEAM |
| 03 May 2022 | ILALABAN PO NAMIN KAYO AT ANG PILIPINAS!<br>@kikopangilinan @lenirobredo #LeniKikoAllTheWay2022<br>#KulayRosasAngBukas #Halalan2022 |

Due to the enormity of the dataset, both English and Filipino/Tagalog tweets were mixed together as we have noticed that most tweets were formed in both languages or what is known as *Tag-lish* or a mix of both Filipino/Tagalog and English. Therefore, all tweets will be processed as such.

*3.2. Data Annotation*

After the data were collected, the researchers worked on annotating most of the tweets with the aim of classifying the tweets into three polarities: positive, neutral, and negative. Positive polarity means that tweets were highly enthusiastic about a certain campaign or candidate. Neutral polarity refers to tweets that have neither agreed nor disagreed with a certain campaign or candidate or even vaguely worded tweets. While negative polarity entails a negative reaction, argument, or displeasure against a candidate, campaign, or both. Table 3 below provides examples to tweets with the above given polarities.

**Table 3.** Some examples of the tweets annotated by hand.

| Polarity | Tweet |
|---|---|
| Positive | Breaking News<br>Bong Bong Marcos to hold rally at Philippine Arena, a good political tactic to potentially be endorsed by Iglesia in 2022. |
| | You nailed it soon to be President of the Philippines Ferdinand "Bong-Bong" Marcos.<br>#SMNIpresidentialdebate |
| | WOMEN CAN LEAD<br>#LetLeniKikoLead2022<br>#LetLeniLead<br>#KulayRosasAngBukas |
| | @trsrhaul GOBYERNONG TAPAT, ANGAT BUHAY LAHAT<br>#LeniKiko2022<br>#IpanaloNa10ParaSaLahat<br>#AbanteBabae |
| Neutral | Bong bong marcos Leni robredo double filter success |
| | ICYMI: Last April 28, supporters of presidential candidates Leni Robredo and Ferdinand "Bongbong" Marcos Jr. locked horns inside Power Plant Mall in Rockwell Center, Makati. The reason? Read: \| via @philstarlife |
| | @iMthinkingPinoy @bongbongmarcos @indaysara @srsasot @smninews @manilabulletin |
| | @JervisManahan @lenirobredo @ABSCBNNews |
| Negative | @manilabulletin Bong Bong Marcos leading na naman daw.<br>That's another f*cking lie.<br>#MarcosMagnanakaw |
| | Bakit si Imelda wala sa mga rallies ni BONG BONG MARCOS?<br>#babakoutmuli |
| | @WinwinEklabu dream on KIKO PANGILINAN!! |
| | Good morning sa lahat, except kay Leni Robredo na hindi pa rin sumusunod sa comelec rules.<br>#LeniTangaSaLahat |

Most of the tweets were annotated by the researchers who know Filipino/Tagalog as some of the tweets were expressed in said language. Similarly, to address biases, the annotators disclosed whether they supported a certain candidate of which they were tasked to only annotate the campaign or candidate they support. In the Table 4 is the summary of the collected tweets that were annotated, subdivided into the campaign they pertain to including the polarity, count, and total tweets.

**Table 4.** Summary of tweets annotated.

| Campaign | Polarity | Tweet Count |
|---|---|---|
| Marcos | Positive | 752 |
| | Neutral | 426 |
| | Negative | 21,506 |
| | **Total** | **22,670** |
| Robredo | Positive | 13,587 |
| | Neutral | 2589 |
| | Negative | 76,015 |
| | **Total** | **92,192** |
| **Total Tweets Annotated** | | **114,851** |

### 3.3. Data Preparation and Pre-Processing

Natural Language Processing techniques highlight data cleaning as one of the central phases to obtain a high accuracy rate. This is also vital for the computer to understand what the data are about [22]. In this phase, the researchers utilized Python 3.8 using IDLEx for the preparation and pre-processing. Using available libraries such as *pandas*, *numpy*, *nltk*, *scipy*, *scikit-learn*, and other libraries, below are each of the steps performed in order for the dataset to be ready for training and testing with the model with the goal of having a high accuracy rate.

#### 3.3.1. Tag Replacement

After the data have been loaded into a *pandas.DataFrame*, the first step was to remove characters such as hashtags (#), question marks (?), exclamation points (!), mention tags (@), enter tags (\n), some emojis and special characters, and even link tags (http/s). These were immediately replaced with the corresponding words such as "hashtag" for the character "#". The same process was performed for the other tags.

#### 3.3.2. Case Transformation

To reduce the discrepancies between lower, upper, and capitalized cases, which can ultimately be represented as different words in the vector space [23], the researchers decided that all cases were to be transformed to lower case.

#### 3.3.3. Tokenization

Using the Natural Language Toolkit (NTLK) library, the researchers used the *TweetTokenizer* as opposed to the normal Tokenizer since we are dealing with tweets. The difference in which is for group of characters such as "<3", it will be tokenized together instead of tokenizing it separately as "<" and "3" [24].

#### 3.3.4. Token Filtering by Length

Once the process of tokenization was conducted, the researchers filtered the words having only four (4) to twenty-five (25) characters in length per word.

#### 3.3.5. Lemmatization

Instead of stemming, the researchers used NTLK's *WordNetLemmatizer* in order to apply morphological analysis to words. For example, the lemma of "was" is "be" and the lemma of "mice" is "mouse" [25].

#### 3.3.6. Stop Word Removal

This process involved using NTLK's updated English stop word corpus to remove English stop words such as "the", "a", "an", "with", "of", etc. In addition, to filter out Tagalog stop words, we also retrieved a Tagalog stop word corpus and removed stop words such as "ang", "at", "kay", "na", "o", "din", "ba", etc.

#### 3.3.7. n-Grams Generation

The term n-Grams is formally defined as "a contiguous sequence of n items from a given sample of text" [26]. Thereby, given any text or word, we can split those into a list of unigrams (1-Gram), bigrams (2-Gram), trigrams (3-Gram), etc.

Using Sci-Kit Learn's *CountVectorizer*, the tokenized data were split into 1-, 2-, and 3-Grams [27]. To obtain the best results, we trained using the *SGDClassifier* in order to obtain the best results as seen in Table 5 below.

**Table 5.** Using the *SGDClassifier*, the data were trained and validated to obtain the best n-Gram value.

| n-Gram | Training Score | Validation Score |
|---|---|---|
| Unigram (1-Gram) | 0.94 | 0.82 |
| Bigram (2-Gram) | 0.99 | 0.82 |
| **Trigram (3-Gram)** | **1.00** | **0.82** |

With the results above, the Trigram (3-Gram) obtained the best training score and will be used for the next phase or process.

### 3.3.8. Training and Testing Data Segmentation

Using SKLearn's train_test_split function, using the *Tag-lish* tweets, an 80–20 split was performed for training and testing, respectively.

### 3.4. Word Embedding Using TF-IDF

The term TF-IDF means Term Frequency times Inverse Document Frequency, a common term weighting scheme in information retrieval, that has also found good use in document classification [28]. Sci-Kit Learn uses the below formula to compute for the *TF* value.

$$TF\ (T, D) = T/D \tag{1}$$

It is calculated as the number of times the term appears in a document (denoted by T) divided by the total number of terms in the document (denoted by D). IDF, on the other hand, is calculated as below.

$$IDF(T) = \log\ [(1 + N)/(1 + DF(T))] + 1 \tag{2}$$

*N* is the total number of documents in the document set and DF(T) is the document frequency of T; the document frequency is the number of documents in the document set that contain the term T. The effect of adding "1" to the IDF in the equation above is that terms with zero IDF, i.e., terms that occur in all documents in a training set, will not be entirely ignored. (Note that the IDF formula above differs from the standard textbook notation that defines the IDF as IDF(T) = log [N/(DF(T) + 1)] [28]). Both TF and IDF values are then multiplied to obtain the TF-IDF, as seen in the formula below.

$$TF\text{-}IDF(T, D) = TF(T, D) * IDF(T) \tag{3}$$

Using Sci-Kit Learn's *CountVectorizer* and *TfidTransformer*, the tokenized data were split it into 1-, 2-, and 3-Grams and then transformed into TF-IDF [27]. To obtain the best results, we trained using the *SGDClassifier* in order to obtain the best results, as seen in Table 6 below.

**Table 6.** Using the *SGDClassifier*, the data were trained and validated to obtain the best TF-IDF transformed and n-Gram value.

| n-Gram | Training Score | Validation Score |
|---|---|---|
| Unigram (1-Gram) | 0.85 | 0.83 |
| Bigram (2-Gram) | 0.86 | 0.82 |
| **Trigram (3-Gram)** | **0.86** | **0.81** |

With the results above, the Trigram (3-Gram) received the best training and validation score and will be used for the next phase or process.

### 3.5. Hyperparameter Tuning

To obtain the best configuration of the Multinomial Naïve Bayes, the researchers used *GridSearchCV* from Sci-Kit Learn's *model_selection* package. Through the use of this function,

it generated the best parameters for the model validated by a 10-fold cross validation. By using a 0.4 alpha parameter, it yielded the highest testing score of 0.839705. The top 10 values are shown in Table 7 below.

**Table 7.** Using the *GridSearchCV*, the model with the top 10 *alpha* parameters yields the following accuracy scores and are ranked accordingly.

| Alpha | Test Score | Ranking |
|---|---|---|
| **0.4** | **0.839705** | **1** |
| 0.5 | 0.838795 | 2 |
| 0.3 | 0.838156 | 3 |
| 0.6 | 0.838102 | 4 |
| 0.7 | 0.837476 | 5 |
| 0.8 | 0.837109 | 6 |
| 0.9 | 0.836770 | 7 |
| 1.0 | 0.836362 | 8 |
| 0.2 | 0.804504 | 9 |
| 0.1 | 0.665983 | 10 |

*3.6. Semi-Supervised Learning*

The semi-supervised learning approach used by the researchers involved the use of Sci-Kit Learn's Self-Training Classifier which requires a base classifier parameter. Using the previously quoted related studies, the researchers chose the Multinomial Naïve Bayes model.

The Multinomial Naïve Bayes model implements the Naïve Bayes algorithm for multinomially distributed data and is one of the two classic Naïve Bayes variants used in text classification [29]. Multinomial Naive Bayes is a probabilistic learning method where the probability of a document *d* being in class *c* is computed as:

$$P(c|d) \propto P(c) \prod_{1 \leq k \leq n_d} P(t_k|c) \tag{4}$$

where $P(t_k | c)$ is the conditional probability of term $t_k$ occurring in a document of class *c*. We interpret $P(t_k | c)$ as a measure of how much evidence $t_k$ contributes that *c* is the correct class. *P(c)* is the prior probability of a document occurring in class *c*. If a document's terms do not provide clear evidence for one class versus another, we choose the one that has a higher prior probability.

In text classification, our goal is to find the best class for the document. The best class in Naïve Bayes classification is the most likely or maximum a posteriori (MAP) class $c_{map}$:

$$c_{map} = argmax_{c \in C} \hat{P}(c|d) = argmax_{c \in C} \hat{P}(c) \prod_{1 \leq k \leq n_d} \hat{P}(t_k|c) \tag{5}$$

We write $\hat{P}$ for *P* because we do not know the true values of the parameters *P(c)* and $P(t_k | c)$, but estimate them from the training set as we will see in a moment.

In Equation (3), many conditional probabilities are multiplied, one for each position $1 \leq k \leq n_d$. This can result in a floating-point underflow. It is, therefore, better to perform the computation by adding logarithms of probabilities instead of multiplying probabilities. The class with the highest log probability score is still the most probable: $\log(xy) = \log(x) + \log(y)$ and the logarithm function is monotonic. Hence, the maximization that is actually performed in most implementations of Naïve Bayes is:

$$c_{map} = argmax_{c \in C} \left[ \log \hat{P}(c) + \sum_{1 \leq k \leq n_d} \log \hat{P}(t_k|c) \right] \tag{6}$$

We first try the maximum likelihood estimate, which is simply the relative frequency and corresponds to the most likely value of each parameter, given the training data. For the priors, this estimate is:

$$\hat{P}(c) = \frac{N_c}{N} \tag{7}$$

where $N_c$ is the number of documents in class $c$, and $N$ is the total number of documents. We estimate the conditional probability $\hat{P}(t_k|c)$ as the relative frequency of term $t$ in documents belonging to class $c$:

$$\hat{P}(t_k|c) = \frac{T_{ct}}{\sum_{t' \in V} T_{ct'}} \tag{8}$$

where $T_{ct}$ is the number of occurrences of $t$ in training documents from class $c$, including multiple occurrences of a term in a document [30].

### 3.7. Performance Evaluation

The researchers applied Sci-Kit Learn's *RepeatedStratifiedKFold* which repeated the validation K times and produced different splits with different randomization in each repetition. Using ten (10) folds with three (3) splits and a random state, the mean accuracy was 0.8403 and standard deviation was 0.0007.

### 4. Results and Discussion

After data preparation and pre-processing, the total number of tweets left were 114,851. With 96,365 (or 83.90%) negative polarity tweets, 15,498 (or 13.49%) and 2988 (or 2.60%) positive and neutral tweets, respectively. A visualization of the count can be seen in Figure 3 below.

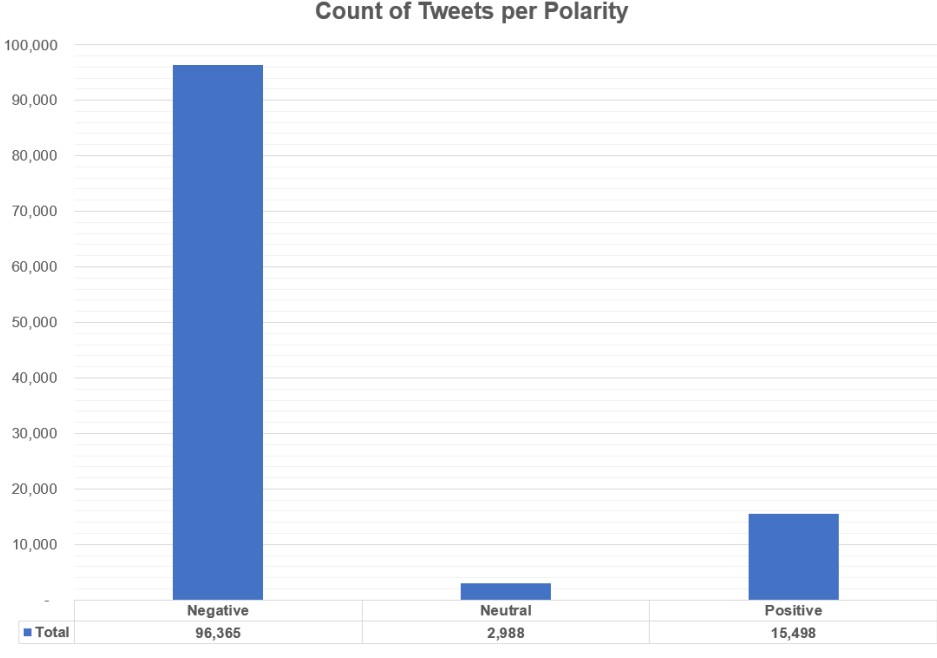

**Figure 3.** Count of the tweets according to polarity whether negative, neutral, or positive.

The annotated data were then divided into training and testing data with a 70–30 split with the 30% testing data having an unlabeled dataset. After training, an accuracy rate of 84.83% was attained by the Multinomial Naïve Bayes model. Seen in Table 8 below is the Confusion Matrix and other metrics.

**Table 8.** The Confusion Matrix and other metrics from the training dataset using the model.

| Label | Predicted Positive | Predicted Neutral | Predicted Negative | Class Recall |
|---|---|---|---|---|
| True Positive | **1187** | 0 | 13,454 | 8.11% |
| True Neutral | 2 | **377** | 2456 | 13.30% |
| True Negative | 31 | 0 | **87,604** | 99.97% |
| Class Precision | 97.30% | 100.00% | 84.63% | |

In addition, below are the word count segmented by polarity in Table 9 and the word cloud in Figure 4.

**Table 9.** The word frequency from the training dataset per polarity.

| Positive | | Neutral | | Negative | |
|---|---|---|---|---|---|
| Word | Count | Word | Count | Word | Count |
| lenirobredo | 8579 | lenirobredo | 1770 | lenirobredo | 47,926 |
| leni | 8613 | abscbnnews | 1284 | leni | 29,432 |
| robredo | 4299 | leni | 970 | robredo | 22,153 |
| 2022 | 2717 | jervismanahan | 843 | 2022 | 15,995 |
| president | 2104 | robredo | 745 | president | 12,964 |
| kiko | 1715 | rapplerdotcom | 367 | bongbongmarcos | 11,628 |
| kikopangilinan | 1392 | inquirerdotnet | 316 | kiko | 10,979 |
| pangilinan | 1380 | cnnphilippines | 272 | pangilinan | 9530 |
| marcos | 1327 | kikopangilinan | 269 | kikopangilinan | 8990 |
| lenikiko | 1118 | president | 259 | marcos | 7036 |

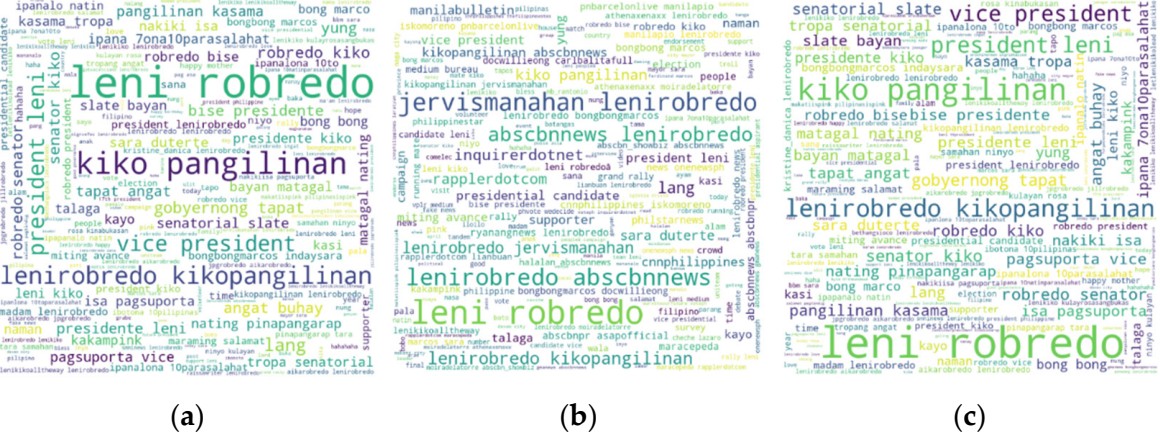

(a) (b) (c)

**Figure 4.** Generated word cloud representing the following polarities: (**a**) positive, (**b**) neutral, and (**c**) negative. The word cloud displays the words used in the dataset; the more frequently the word was used, the bigger it is displayed in the word cloud.

## 5. Conclusions

With the aim of acquiring the general sentiment of Filipinos during the 2022 Philippine Presidential Elections, it has no doubt been a divisive one. Many families and friendships have been shattered especially based on whom they support. By this research, we have confirmed these sentiments, having 83.90% of the tweets sent out that are negative in nature, and it did not matter whether the tweet is for or against a certain candidate. This is followed by only 13.49% and 2.60% positive and neutral tweets. By using Natural Language Processing techniques, data were collected, prepared, and pre-processed. Using hyperparameter tuning, the best possible parameter was used to train the base classifier, the Multinomial Naïve Bayes model, which was then used as a parameter for the Self-Training

model for the semi-supervised learning approach with 30% unlabeled data, an accuracy rate of 84.83% higher than the researchers' previous study's accuracy rate.

Though it was expected that this election was as divisive as it was predicted, the researchers hoped that after the election, Filipinos would come together and support around the majority decision with the election of another Marcos in Malacañang. Instead of holding on to crab mentality, a trait in which Filipinos are fond of, we should support each other and lift each other up rather than pull each other down.

**Author Contributions:** Conceptualization, J.J.E.M. and C.N.V.; methodology, J.J.E.M. and C.N.V.; validation, J.J.E.M. and X.A.I.; formal analysis, J.J.E.M. and C.N.V.; writing—original draft preparation, J.J.E.M.; writing—review and editing, C.N.V., X.A.I. and J.-H.J.; visualization, J.J.E.M.; supervision, J.-H.J.; project administration, J.-H.J.; All authors have read and agreed to the published version of the manuscript.

**Funding:** This research received no external funding.

**Data Availability Statement:** Not applicable.

**Acknowledgments:** The authors wish to thank I-Shou University and Taiwan's Ministry of Education. This work was supported in part as the corresponding author is a scholar of Taiwan's Ministry of Education under the New Southbound Elite Research and Development Degree Program. The authors would also like to thank the ALMIGHTY GOD for His guidance from the start until the completion of this study.

**Conflicts of Interest:** The authors declare no conflict of interest.

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
