# Peer review of "A Semi-Supervised Approach to Sentiment Analysis of Tweets during the 2022 Philippine Presidential Election"

_information, doi:10.3390/info13100484_

Round 1

Reviewer 1 Report (Previous Reviewer 2)

This is an interesting and well-written paper. 

The most interesting for me is the use of two languages English and actually Filipino / Tagalog. Please give some more details in the methodology section about this mix of languages and how have you dealt with this mix.  For example, how many tweets are in English, and how many are in Filipino / Tagalog? How have you created the training and testing data sets in regard to the two languages?  Is the general sentiment of Filipinos expressed in English and in Filipino / Tagalog the same?

Author Response

Good Day Reviewer,

Due to the enormity of our data (about 114,851), we cannot specify the amount of English and Filipino/Tagalog tweets, however, the most Filipino tweets are in Tag-lish or a mix of both English and Filipino/Tagalog. Therefore, all tweets were processed as such. For the training and testing data, we used the 80-20 random split as recommended. Also, since these were mixed, the general sentiment remains the same.

The reason why Filipinos tweet in both languages is that English is the Philippines' second (some may even say first) language. Therefore, most people would be able to understand both languages.

I have edited the manuscript as attached. Hope that the revisions are enough. 

Thank you!

Reviewer 2 Report (Previous Reviewer 1)

Dear Authors,

There is not any serious hindrance before publication. Congratulations, the paper has had thoughtful upgrades since the first version.

Author Response

Thank you very much for your comment! Your comment has greatly improved our paper. 

Round 2

Reviewer 1 Report (Previous Reviewer 2)

Thank you for your answer. 

This manuscript is a resubmission of an earlier submission. The following is a list of the peer review reports and author responses from that submission.

Round 1

Reviewer 1 Report

Thank you for inviting me to evaluate the article titled “Twitter Sentiment Analysis of the 2022 Philippine Presidential Elections using Multinomial Naïve Bayes”.The article presents multinomial Naive Bayes’ classification of a political event. Although the data would be a useful for a more deeper analytics, e.g. Kovács, T.; Kovács-GyÅ‘ri, A.; Resch, B. 2021) in present form it would not be suggested for publication, as it suffers from formal and methodological mistakes.

Methodologically, the authors use a well-known and widely used approach of sklearn MultinomialNB classifier which was trained with manually labeled tweet data. It uses conditional probabilities of each lexical feature occurring in either positive, negative, or neutral text in the training data. Combining with ngrams the accuracy is better (since the early 2010, see Cramer and Gordon). Considering statistical analytics, besides SVN this method offers the best performance. However, the SOTA would force us to train and apply a transformer based approach to transfer learning that may ignore manually labeling.

Methodically speaking, it is problematic that the article did not clarify the followings:

- The method is highly reliant on priors, so the training data has to be representative. Overall, 31517 tweets were annotated (which is the full dataset!) then the authors used 80/20 split, and made their conclusion about the sentiment during the presidential election...

- The distribution of the processed tweets. We do not know how widely this platform was used in the analyzed country.

- We do not know how many tweets of natural person were used. The appearance of the word “Breaking News” shows that news bots or retweets were also among the processed tweets.

- Does the distribution of individual users and tweets follow a certain pattern?

- Which rules were followed by the annotators when they labeled some tweets as neutral? According to the article, they assigned this task to researchers which supports the chance of misclassification (which is still subjective with no strict classification manual)

- Academic editing is required.

Please consider a more detailed introduction and description of how you handled and preprocessed your data.

Cited articles:

- AllforJan: How Twitter Users in Europe Reacted to the Murder of Ján Kuciak—Revealing Spatiotemporal Patterns through Sentiment Analysis and Topic Modeling https://www.mdpi.com/2220-9964/10/9/585

- Improvement of a Naive Bayes Sentiment Classifier Using MRS-Based Features https://aclanthology.org/S14-1003.pdf

- Blog: https://towardsdatascience.com/sentiment-analysis-comparing-3-common-approaches-naive-bayes-lstm-and-vader-ab561f834f89

- Blog 2: https://github.com/richlipkin/Language-Processing-N-Grams/blob/master/Language%20Detection-2.ipynb?utm_source=pocket_mylist

Reviewer 2 Report

The authors investigate a topic that is not largely studied, thus the paper is interesting and it is quite well written.

However, I believe that the literature review is not complete. Many studies exist that investigate sentiment analysis during elections. I think that the authors have to make a sub-section and discuss these papers and the approaches they have used.

For example

Bansal, B., & Srivastava, S. (2018). On predicting elections with hybrid topic based sentiment analysis of tweets. Procedia Computer Science, 135, 346-353.

Rodríguez-Ibáñez, M., Gimeno-Blanes, F. J., Cuenca-Jiménez, P. M., Soguero-Ruiz, C., & Rojo-Álvarez, J. L. (2021). Sentiment analysis of political tweets from the 2019 Spanish elections. IEEE Access, 9, 101847-101862.

Sharma, A., & Ghose, U. (2020). Sentimental analysis of twitter data with respect to general elections in india. Procedia Computer Science, 173, 325-334.

Yaqub, U., Malik, M. A., & Zaman, S. (2020, November). Sentiment analysis of Russian IRA troll messages on Twitter during US presidential elections of 2016. In 2020 7th International Conference on Behavioural and Social Computing (BESC) (pp. 1-6). IEEE.

Bansal, B., & Srivastava, S. (2019). Lexicon-based Twitter sentiment analysis for vote share prediction using emoji and N-gram features. International Journal of Web Based Communities, 15(1), 85-99.

Praciano, B. J. G., da Costa, J. P. C. L., Maranhão, J. P. A., de Mendonça, F. L. L., de Sousa Júnior, R. T., & Prettz, J. B. (2018, November). Spatio-temporal trend analysis of the Brazilian elections based on Twitter data. In 2018 IEEE International Conference on Data Mining Workshops (ICDMW) (pp. 1355-1360). IEEE.

and many more.

Please explain why you selected to use Naïve Bayes and not Support Vector Machine or Random Forest or any other method.

What are the limitations and the implication of your study?

At the end of the introduction please add the structure of the paper.